# Autonomous RFID Sensor Node Using a Single ISM Band for Both Wireless Power Transfer and Data Communication

**DOI:** 10.3390/s19153330

**Published:** 2019-07-29

**Authors:** Abderrahim Okba, Dominique Henry, Alexandru Takacs, Hervé Aubert

**Affiliations:** LAAS-CNRS, Université de Toulouse, CNRS, INSA, UPS, 31400 Toulouse, France

**Keywords:** autonomous wireless sensor nodes, wireless power transfer, rectenna, RFID sensor tag

## Abstract

This paper addresses the implementation of autonomous radiofrequency identification sensor nodes based on wireless power transfer. For size reduction, a switching method is proposed in order to use the same frequency band for both supplying power to the nodes and wirelessly transmitting the nodes’ data. A rectenna harvests the electromagnetic energy delivered by the dedicated radiofrequency source for charging a few-mF supercapacitor. For supercapacitors of 7 mF, it is shown that the proposed autonomous sensor nodes were able to wirelessly communicate with the reader at 868 MHz for 10 min without interruption for a tag-to-reader separation distance of 1 meter. This result was obtained from effective radiated powers of 2 W during the supercapacitor charging and of 100 mW during the wireless data communication.

## 1. Introduction

Recently, due to the development of many Internet of Things (IoT) [1] applications, radiofrequency (RF) power transmission has garnered particular interest for the implementation of battery-free and/or energy autonomous wireless sensors nodes (WSNs). Two scenarios can be defined: (*i*) wireless power transfer (WPT), in which a dedicated source provides electromagnetic energy and a rectenna is used to harvest the energy; and (*ii*) Energy Harvesting (EH), where electromagnetic energy is harvested from the electromagnetic radiation of RF sources available in the environment. These two scenarios can be used for many applications, such as structural health monitoring [2] and automotive health monitoring [3]. In practice, the WPT scenario has a greater chance of success due to the very low amount of RF energy usually available in the environment.

At present, a great number of radiofrequency identification (RFID) systems have been developed and used in many areas. Some applications use passive tags [4,5], while others use two different frequency bands for receiving RF power and transmitting data, respectively [6,7]. Some works pay particular attention to the readers and the RF communication issues. For example, in [8] an adaptive power control is proposed to counteract collision issues in dense reader networks. In this paper, the problem of collision between readers is highlighted and the implementation of a distributed adaptive power control (DAPC) algorithm is presented. Other works focus on the RFID tag, as in [9] where an RFID passive tag including sensors was carried out, the reader was used as the main energy source to supply the tag and the sensors, and an optional photovoltaic cell was integrated as a second energy harvester. 

In the same context, this paper addresses the implementation of a new batteryless autonomous wireless RFID tag integrating sensors supplied using wireless power transfer. The RFID tag used in this work was originally battery-assisted. However, the battery was replaced by a rectifier connected to a power management unit and a supercapacitor to give more autonomy to the overall system. For size reduction, the same frequency band is used here for both supplying power to the node and wirelessly transmitting the node’s data. The novelty of this work is that a switching protocol between RF energy harvesting and data transmission is proposed to allow overcoming the RF collisions between the RF source and the reader and obtaining an energy-efficient system. In a previous work published by some of us [10], the same frequency was used for WPT (downlink from RF source to wireless nodes) and the Long Range (LoRa) wireless communication (uplink from nodes to the LoRa gateway). However, the short RF source-to-node separation distances (typically a few meters) and the very long range of the LoRa wireless communication (more than 1 km) make it impossible to locate the RF source and the gateway in the same place. Moreover, even if a node’s supercapacitor is charged, the RF source in [10] continuously transmits RF power, which poses a critical issue of the energy efficiency of the wireless system. For battery-assisted RFID technologies, the distance between the sensor tags and the reader is in on the order of magnitude of the RFID communication range (i.e., a few meters [11]) and consequently, we propose here to locate the WPT sources and RFID reader in the same place. Moreover, for the first time, a switching protocol between WPT and RFID wireless communication is reported for carrying out an energy-efficient integrated wireless system. It includes a cold-start to charge the supercapacitor and switches to alternate between wireless communication and WPT. The tag used here is a multipurpose Electronic Product Code (EPC) class 1 and class 3 compliant on-the-shelf RFID device working at 868 MHz (AMS SL900A [12]). It integrates an internal temperature sensor and may eventually incorporate two external sensors. In this work, we observed that the tag to reader distance could reach 4 to 5 meters when the RFID tags did not include active sensors. However, this distance was reduced to 1 meter when the sensors were used. The proposed system is thus dedicated to short-range sensing applications.

The paper is organized as follows: In Section 2, the topology of the overall system is detailed (Section 2.1), and the WPT/data switching method (Section 2.2) is proposed. In Section 3, the experimental setup is described (Section 3.1), and the obtained measurement results using the proposed autonomous wireless sensor node (Section 3.2) are discussed.

## 2. WPT/Data Transmission Switching Method

### 2.1. Topology of the Wireless Setup

The overall wireless system combining WPT and wireless data communication operating in the same ISM (industrial, scientific, and medical) frequency band is illustrated in Figure 1. It is composed of:
(i)The WPT transmitter, that is, the RF (ISM band) source connected to the transmitting antenna. The effective radiated power (ERP) here is lower than 2 W for compliance with electromagnetic exposure limitations/standards [13].(ii)The WPT receiver, composed of the rectenna for harvesting the incident electromagnetic energy, a power management unit (PMU) with a DC-to-DC boost converter (e.g., the Bq25504 device [14]) for delivering the voltage required by the RFID tag for nominal operation, and the supercapacitor.(iii)The sensor data transceiver (reader and antenna).(iv)The batteryless autonomous RFID tag used to perform the wireless transmission of the node’s data.

The proposed wireless sensor node alternatively performs WPT and RFID communication. During the WPT illustrated in Figure 1a, the single-pole double-throw (SPDT) switches SW1 and SW2 are simultaneously connected to port 2, whereas the single-pole single-throw (SPST) switch SW3 is in the OFF state. The supercapacitor is then wirelessly charged by means of the RF power harvested by the rectenna and the PMU from the far-field WPT approach. During the wireless communication of data illustrated in Figure 1b, the SPDT switches SW1 and SW2 are simultaneously connected to port 1, while the SPST SW3 is in the ON state. The RFID sensor tag is then supplied power by the supercapacitor, and the RFID reader is able to transmit frames and receive responses from the RFID sensor.

### 2.2. Switching Method for Alternatively Supplying Power to the RFID Tag and Transmitting the Sensor Data to the Reader

In order to highlight the benefits of alternating WPT and wireless data communication using the same ISM band, the following experiment was performed. The RFID reader periodically transmitted a set of RFID frames, as detailed in Section 3, while the following steps were sequentially applied:

Step i:The 7 mF supercapacitor was initially discharged and the voltage at its port was 0 V. SW2 was connected to port 2 and SW3 was OFF;Step ii:The reader and RF source (ERP < 2 W) were both activated. Far-field WPT began when the required DC voltage and DC power were available at the PMU input, the latter was activated and proceeded to the so-called cold start-up procedure. In this step, SW2 was connected to port 2 and SW3 was OFF;Step iii:The cold start-up procedure continued, and the supercapacitor charged faster than in step ii. SW3 was OFF;Step iv:When the supercapacitor was able to deliver a voltage of at least 3 V to the RFID tag, the RF source was shut down, and the RFID sensor tag was ready to wirelessly transmit its data to the reader. SW2 was then connected to port 1 and SW3 was ON;Step v:During the wireless data communication, the supercapacitor discharged until the supplied voltage took the predefined voltage threshold, denoted by V_Min_ (here, V_Min_ = 2.4 V)_._ In this step, SW2 was connected to port 1 and SW3 was ON;Step vi:When the threshold V_Min_ was reached, the RF source was switched on in order to wirelessly charge the supercapacitor. SW2 was connected to port 2 and SW3 was OFF.

Steps iv to vi were repeated as long as the wireless data transmission was required.

## 3. Experimental Results

### 3.1. Experimental Setup

A realistic scenario where WPT is used for supplying power to the autonomous RFID sensor node is considered here. This experiment was carried out as a proof of concept of the switching method described in Section 2.2. The microwave generator MG3694B from Anritsu was used to inject the RF signal at the input of the transmitting (Tx) Patch antenna (860–885 MHz) via a coaxial line. The rectenna was placed at 1 m in line of sight, and harvested the radiated electromagnetic energy. The experimental setup is shown in Figure 2a.

The rectenna is shown in Figure 2b. It is composed of a flat dipole antenna and a half-wave rectifier and covered the entire ISM 868 MHz band. When the harvested DC voltage and DC power were sufficient at the PMU input, the supercapacitor started charging. When the charge was complete, the microwave generator was manually switched off for this proof-of-concept. The tag was then ready to transmit its data to the reader, if requested. The experimental setup was placed in an anechoic chamber to prevent eventual electromagnetic interferences or multipath effects.

The design of the rectenna used in this work is detailed in [15]. The antenna and the rectifier were overlapped, allowing a planar compact energy harvesting structure (about 10.5 × 6 cm^2^) to be achieved. The metallic layer of one monopole is in mechanical and electrical contact with the ground plane of the rectifier. A very short wire was used to connect the other monopole with the rectifier input. This rectenna allows the PMU to be activated for low incident power densities. As can be seen in Figure 3a, the rectenna allowed for the harvesting of a DC voltage of 470 mV at 860 MHz with an incident power density of 0.43 µW/cm². The corresponding DC power is about 22.2 µW. Figure 3b depicts the rectenna DC power and efficiency as a function of the incident power density. The 10 µW required to turn on the PMU was reached when the illuminating power density was 0.26 µW/cm² at 860 MHz. For the electromagnetic power density of 2.1 µW /cm², the rectenna efficiency exceeded 25% and reached 37% at 860 MHz with a load of 10 kΩ. Even though this load is different from the input impedance presented by the Bq25504 that varied according to the input voltage, this result gives good insight into the rectenna’s performance. In practice, this low-profile rectenna was able to turn on the PMU “Bq25504” with low incident power densities in the ISM 868 MHz band.

### 3.2. Measurement Results and Discussion

During a period of 10 seconds, the set of four consecutive RFID frames are wirelessly transmitted to the tag. The first frame asks for the tag’s EPC and received signal strength indicator (RSSI). The second and third frames ask for the temperature measured by the internal node sensor and the data of the external photoresistive sensor, respectively. The last frame asks for the supply voltage of the RFID tag.

Figure 4 reports the voltage at the supercapacitor port for three different ERPs: 33 dBm (2W), 31 dBm, and 29 dBm. The resulting power densities at the rectenna location (1 m in front of the reader) were then 12.6, 7.9, and 5 µW/cm², respectively. The cold start-up charging times (t_c_) were 6, 8, and 12 min, respectively, for a total initial charging time of 8, 10, and 15 min, respectively. As expected, the higher the power density, the shorter the charging time. Consequently, the proper estimation of the power density is necessary to predict the delay time from which the tag is able to start the wireless transmission of data. The data transmission along with the voltage measured at the port of the supercapacitor are given in Figure 5a for the power density of 12.6 µW/cm². The different steps described in Section 2 are easily observable (identified by the roman numerals i, ii, iii, iv, v, and vi in Figure 5a). For steps i and ii, the system was initialized and both the RF source and RFID reader were activated. 

Success or failure in the wireless data transmission is indicated by the binary function (black curve) at the bottom of Figure 5a: when this function equals 1, the sensor tag is detected and all the four frames (RSSI, temperature, luminosity, and supply voltage) are successfully transmitted by the RFID sensor tag; when the function equals 0, the wireless data transmission do not (or do partially) occur. As expected, it can be observed that some data were not successfully transmitted when the RFID reader and the RF source worked simultaneously (step iii to step iv after the initial charging). Indeed, when the RFID reader and the RF source worked simultaneously, high-power harmonics (>29 dBm) radiating from the RF source caused the reduction of the signal-to-noise ratio. As a consequence, the RFID communication was disturbed since it operated at the same frequency but with lower radiated power (typically lower than 20 dBm). Some data were thus unsuccessfully transmitted. Alternating WPT and RFID communication, as proposed in this work, is thus necessary when using a single ISM band for the wireless power supply and communication with the node. 

The supplied voltage, which is measured by the RFID sensor tag, is shown by black triangles in Figure 5a. The successfully transmitted data (RSSI, temperature, and resistance of the photoresistor) are reported in the chronogram of Figure 5b. When the supercapacitor was fully charged (step iv), the RF source was turned off, the supercapacitor started discharging, and the data transmission was therefore possible, as shown by the binary function (black curve) which equaled 1 between t_1_ = 15 min and t_2_ = 25 min. The RFID sensor tag was then able to transmit four consecutive frames containing the data from the sensor node with the period of 10 seconds. When the voltage at the supercapacitor port was lower than V_min_ (2.4 V), the RFID sensor tag could not wirelessly transmit its data. Consequently, the RF source was switched on (step vi) in order to recharge the supercapacitor. The turn on/turn off cycle of the RF source used for WPT was repeated as long as data transmission was required. 

In this work, the sensor tag was autonomous for 10 min with a 7 mF supercapacitor. Key parameters that must be considered to predict such autonomy duration are: (a) the capacity of the supercapacitor, (b) the power consumption of the sensor tag, and (c) the duration T between data transmission. Table 1 summarizes the obtained WPT performances. 

## 4. Conclusions

An autonomous RFID sensor node using wireless power transfer was reported. The WPT and data transmission shared the same ISM 868 MHz band, and an original switching solution between WPT and data transmission was experimentally validated. By using WPT instead of a battery, a supercapacitor of 7 mF could be charged in 10 min for an incident electromagnetic power density of 7.9 µW/cm². The proposed RFID sensor tag was able to transmit its data (EPC, RSSI, internal temperature) for 10 min without interruption. Future work will focus on the implementation of the compact sensor node integrating the rectenna, the PMU, and the RFID sensor tag. A self-tuning sensor technique is also investigated for the automatic control of the switches’ states. Many solutions can be proposed for practical implementations, such as applying an automatic switching between WPT and communication data by using preliminary specified time slots for each operation. This simple, but not optimized, solution has a great chance of success, but other solutions could be investigated.

## Figures and Tables

**Figure 1 sensors-19-03330-f001:**
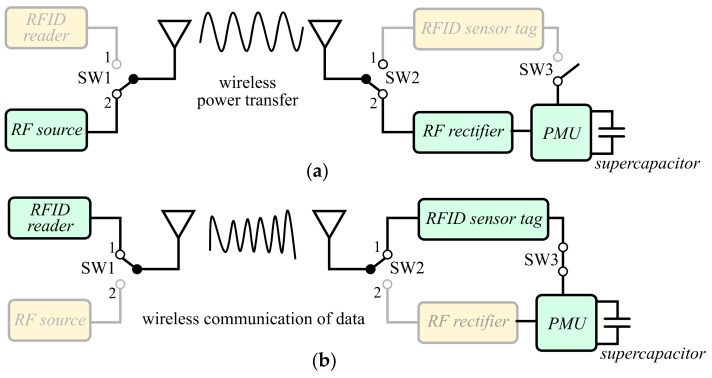
Block diagram of the proposed autonomous wireless sensor node during (**a**) wireless power transfer and (**b**) wireless data communication. SW1 controls the transmitting mode and SW2 and SW3 control the power supply of the radiofrequency identification (RFID) sensor tag. PMU: power management unit.

**Figure 2 sensors-19-03330-f002:**
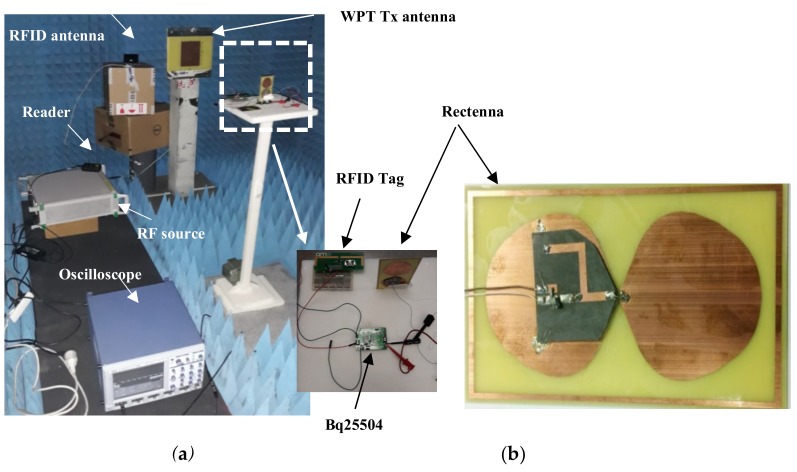
Photographs of (**a**) the experimental setup and (**b**) the node rectenna. Tx: transmitting.

**Figure 3 sensors-19-03330-f003:**
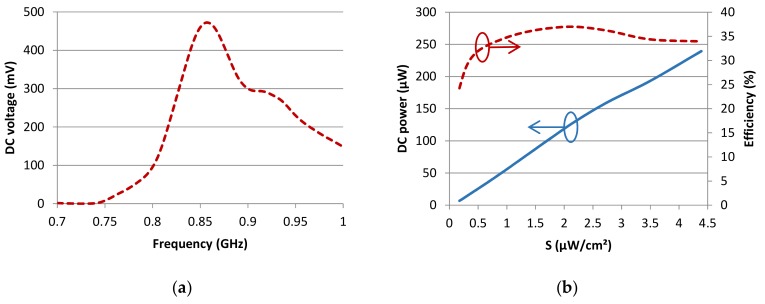
(**a**) Measured DC voltage as a function of the frequency with 10 kΩ load (see [13]); (**b**) Rectenna harvested DC power (continuous curve) and radiofrequency (RF)-to-DC efficiency (dashed curve) at 860 MHz with a 10 kΩ load as a function of the incident RF power density S (from [15]).

**Figure 4 sensors-19-03330-f004:**
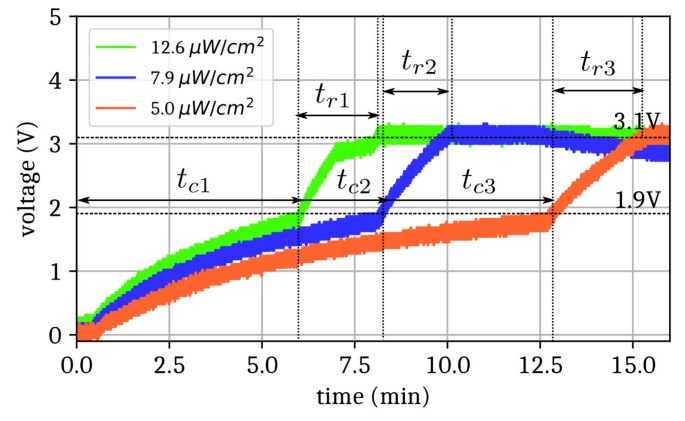
Cold start-up duration (t_c_) and charging time (t_r_) of the supercapacitor (7 mF) of the PMU for three different RF power densities: 12.6 µW/cm² (green), 7.9 µW/cm² (red), and 5.0 µW/cm² (blue) measured with an oscilloscope.

**Figure 5 sensors-19-03330-f005:**
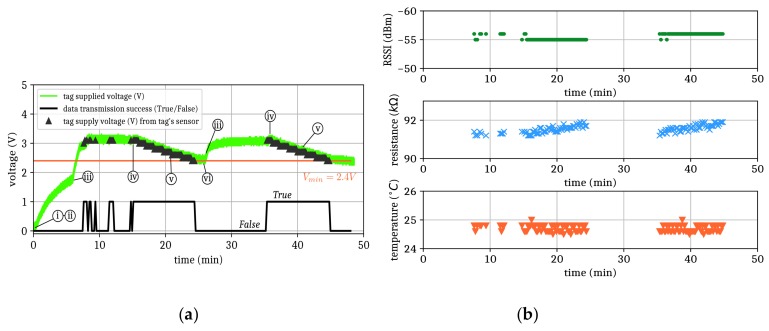
(**a**) Voltage at the port of the supercapacitor measured with oscilloscope (green curve) with RF power density of 12.6 µW/cm²; (**b**) Received signal strength indicator (RSSI) (green dots), resistance of the photoresistor (blue crosses), and temperature (orange downward-pointing triangles) are displayed only when the sensor data were successfully transmitted.

**Table 1 sensors-19-03330-t001:** Cold start-up time and second charging time as a function of the incident power density for the 7-mF supercapacitor.

Effective Radiated Power	Incident Power Density	Measured Cold Start-Up Time (t_c_)	Measured Overall Charging Time (t_r_)
29 dBm	5.0 µW/cm²	12 min	15 min
31 dBm	7.9 µW/cm²	8 min	10 min
33 dBm	12.6 µW/cm²	6 min	8 min

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
