# Peer review of "Autonomous RFID Sensor Node Using a Single ISM Band for Both Wireless Power Transfer and Data Communication"

_sensors, 2019, doi:10.3390/s19153330_

Round 1
Reviewer 1 Report
Although this is a nice work, it does not present new technical improvements or contributions compared to published art for thee following reason.
(1) The first major limitation flaw is the 1 meter measurement distance which is way too short for RFID applications.
(2) The second major limitation is "manually" switching RF source off. I understand that this is proof of concept. Still, the author needs to be able to show the path forward for practical implementation.
(3) What is the reason for "it can be observed that some data are not successfully transmitted when 162 the RFID reader and the RF source work simultaneously" . The author should provide technical reasons.
(4) The efficiency curve shown in Fig.3(b) is typical without optimizing specifically for a specific input power level. The author use 10K ohms as the load for all power levels for this efficiency curve, but the BQ25504 has dynamic MPPT operation which varies input impedance according to different input voltage. So, how useful, truthful, practical is this efficiency curve.
Author Response
Please see the attachement.

Reviewer 2 Report
A switching approach is utilized to charge an RFID node wirelesslessly. Rest of the approach is standard and not novel. Though this appears to be interesting, similar effort has been done by other authors in the literature and the paper authors fail to cite/discuss/contrast the work of others. The main issue is novelty. I suggest the authors to read Haifeng Niu's paper on RFID reader and tag design with the charging aspect. Other papers also appeared which might be of interest to the authors. A benchmarkign study needs to be done to contrast this work and the effort in the literature.
In summary, the proposed effort is interesting but needs additional effort to contrast this work with that of the others.
Author Response
Please see the attachement.

Round 2
Reviewer 1 Report
The revisions author made adds more clarity to the paper and has address my previous comments.
Author Response
No more responses are needed.
Reviewer 2 Report
The authors should cite more relevant literature on the applications of RFID as sensor and its applications.
The major issue is that the paper is not ready as it does not cite relevant literature and show the original contributions contrasting it with the literature
